# Selective gating to vibrational modes through resonant X-ray scattering

Rafael C. Couto[1,2], Vinícius V. Cruz[1], Emelie Ertan[3], Sebastian Eckert[4,5], Mattis Fondell[5], Marcus Dantz[6], Brian Kennedy[5], Thorsten Schmitt[6], Annette Pietzsch[5], Freddy F. Guimarães[2], Hans Ågren[1], Faris Gel'mukhanov[1,7], Michael Odelius[3], Victor Kimberg[1,7] & Alexander Föhlisch[4,5]

The dynamics of fragmentation and vibration of molecular systems with a large number of coupled degrees of freedom are key aspects for understanding chemical reactivity and properties. Here we present a resonant inelastic X-ray scattering (RIXS) study to show how it is possible to break down such a complex multidimensional problem into elementary components. Local multimode nuclear wave packets created by X-ray excitation to different core-excited potential energy surfaces (PESs) will act as spatial gates to selectively probe the particular ground-state vibrational modes and, hence, the PES along these modes. We demonstrate this principle by combining ultra-high resolution RIXS measurements for gas-phase water with state-of-the-art simulations.

[1] Theoretical Chemistry and Biology, School of Biotechnology, Royal Institute of Technology, S-106 91 Stockholm, Sweden. [2] Instituto de Química, Universidade Federal Goiás, Campus Samambaia, CP 131, Goiânia, Goiás 74001-970, Brazil. [3] Department of Physics, Stockholm University, AlbaNova University Center, 10691 Stockholm, Sweden. [4] Institut für Physik and Astronomie, Universität Potsdam, Karl-Liebknecht-Strasse 24-25, 14476 Potsdam, Germany. [5] Institute for Methods and Instrumentation in Synchrotron Radiation Research G-ISRR, Helmholtz-Zentrum Berlin für Materialien and Energie Albert-Einstein-Strasse 15, 12489 Berlin, Germany. [6] Research Department Synchrotron Radiation and Nanotechnology, Paul Scherrer Institut, CH-5232 Villigen PSI, Switzerland. [7] Laboratory for Nonlinear Optics and Spectroscopy, Siberian Federal University, 660041 Krasnoyarsk, Russia. Correspondence and requests for materials should be addressed to F.G. (email: faris@theochem.kth.se) or to M.O. (email: odelius@fysik.su.se) or to A.F. (email: alexander.foehlisch@helmholtz-berlin.de).

Chemical reactions are strongly affected by vibrational excitations through changes in the positions of the nuclei. Vibrational control over photochemical processes can be effectively executed by excitation of vibrational modes spatially aligned along the reaction coordinate. Experimental evidence of vibrationally mediated photochemistry has been reported earlier for isolated molecules and nanocrystals[1–4]. However, an efficient selection of a particular reaction pathway in polyatomic molecules by means of vibrational excitation is a rather difficult task because of the elevated number of coupled degrees of freedom and the high density of vibrational states. Addressing such a challenging objective requires the development of special experimental schemes. Thanks to ultra-high spectral resolution, modern resonant inelastic X-ray scattering (RIXS) spectroscopy provides a unique opportunity for filtering the ground-state vibrations using spatially selective nuclear dynamics in intermediate core-excited states. This spatial selectivity stems from the landscape of the core-excited potential energy surface (PES) that drives the propagation of the multimode wave packet along particular reaction coordinates. We chose for our study the $H_2O$ molecule that constitutes a crucial benchmark system for demonstrating this principle, not only because of its inherent importance for physical chemistry but also for being a basic model for triatomic $AB_2$ molecules.

In the following, we present ultra-high resolution (see Methods) RIXS data that we combine with state-of-the-art *ab initio* electron structure and time-dependent wave packet calculations. By tuning the photon energy in resonance with specific core-excited states we can selectively probe different extended regions of the ground-state potential that correspond to distinct vibrational modes. The main idea of our experiment is to analyse the spatial shape of the electronic ground-state vibrational wave functions from the point of view of nuclear wave packet propagation along state-specific reaction coordinates of the core-excited state. Namely, we consider the three lowest core-exited states of water: the dissociative $\left|O1s^{-1}4a_1^1\right\rangle$ state with the valley of the potentials of the stretching modes along the OH bonds, the $\left|O1s^{-1}2b_2^1\right\rangle$ state with the nuclear wave packet localized between the OH bonds (along the symmetric normal coordinate) and the $\left|O1s^{-1}2b_1^1\right\rangle$ state with the nuclear wave packet primarily excited in the bending mode.

## Results

**Theoretical approach and vibrational analysis.** The vibrational energy levels of the ground electronic state of gas-phase water have been studied by several spectroscopic techniques. The low-lying vibrational states were widely investigated by means of one-photon spectroscopy[5–8], but for reaching higher states, advanced techniques had to be applied that employ a two-photon[9] or three-photon[10] excitation scheme. From the theoretical point of view, these vibrational states can also be obtained using high-level *ab initio* calculations[11–13], but these methods are computationally extremely expensive. It will be shown in this paper that spatially selective nuclear dynamics in core-excited states allows one to study the vibrational levels of ground-state water in a long range along selected reaction coordinates.

In the RIXS simulations, the three vibrational modes of the water molecule are tackled within a 2D + 1D model, where the coupling of the two-dimensional (2D) stretching motion with the one-dimensional (1D) bending mode is neglected (see Supplementary Note 1 for details). The bending potentials are computed around the point of vertical transition. The two coupled stretching motions are treated explicitly by solving the 2D time-dependent Schrödinger equation on the full

bidimensional PESs of the core-excited and final states with the 2D nuclear Hamiltonians $h_c$ and $h_f$ expressed in valence coordinates, respectively. In contrast, the bending motion is treated by solving the time-independent 1D Schrödinger equation and computing the Franck–Condon amplitudes $\langle m_i | m_j \rangle$ between the vibrational sublevels of the electronic states $i$ and $j$. Despite of the strong anharmonic coupling of the normal modes in $H_2O$, it is convenient to assign each vibrational state by three vibrational quantum numbers $(n_s, m, n_a)$, representing the symmetric stretching, bending and antisymmetric stretching normal modes, respectively.

Using a time-dependent representation[14,15], we compute the RIXS cross-section within this model

$$\sigma_{fc}(\omega', \omega) = \sum_{m_f, m_c', m_c} \langle 0 | m_c' \rangle \langle m_c' | m_f \rangle \langle m_f | m_c \rangle \langle m_c | 0 \rangle$$
$$\times \mathrm{Re} \int_0^\infty dt\, e^{i(\omega - \omega' - \omega_{f0} - \varepsilon_{m_f} + \varepsilon^{(0)})t} e^{-i\Gamma_f t} c_{m_c' m_c}(t) \tag{1}$$

as the function of the energy loss $\omega - \omega'$. Here $\omega'$ is the frequency of the scattered photon, $\omega_{f0} = E_f^{min} - E_0^{min}$ is the difference between the minima of the ground- and final-state PES, $\varepsilon^{(0)}$ and $\varepsilon_{m_f}$ are the total zero-point energy of the ground state and the bending vibrational energy of the final state. To find the autocorrelation function

$$c_{m_c' m_c}(t) = \langle \Phi_{m_c'}(0) | e^{-ih_f t} | \Phi_{m_c}(0) \rangle,$$
$$|\Phi_{m_c}(0)\rangle = \int_0^\infty dt_1 e^{-\Gamma t_1} e^{i(\Omega - \varepsilon_{m_c} + \Delta)t_1} |\Psi_c(t_1)\rangle \tag{2}$$

the integrated wave packet $|\Phi_{mc}(0)\rangle$ is defined by the nuclear dynamics in the core-excited state $|\Psi_c(t)\rangle = \exp(-ih_c t)|0\rangle$, where $\Omega = \omega - w_{c0}^{vert} + \varepsilon^{(0)}$ is the detuning of the incoming photon frequency from the frequency $w_{c0}^{vert}$ of the vertical transition $0 \to c$. Here $\Delta = E_c(\mathbf{R}_0) - E_c(\mathbf{R}_0^{(c)})$, $\mathbf{R}_0$ and $\mathbf{R}_0^{(c)}$ are the coordinates of the potential minima of the ground and core-excited three-dimensional (3D) potentials. One should notice that the vibrational progression in general depends on the polarization of incoming and scattered X-rays. This dependence originates from the breakdown of the Born–Oppenheimer approximation. All the core-excited states of the water molecule studied here are nicely isolated and the Born–Oppenheimer approximation is preserved. Hence, the polarization effect is neglected in our simulations based on equation (1) (see Supplementary Note 2 for more details).

A molecule has independent vibrations (normal modes) only in the harmonic approximation. The real molecular potential of the stretching motion of $H_2O$ (Fig. 1c,e) deviates strongly from the elliptic shape of a harmonic potential. Therefore, the stretching modes are not independent anymore but they are coupled because of the anharmonicity (see Supplementary Note 3). The shape of the vibrational wave functions (Fig. 2c–h) demonstrates the complete breakdown of the harmonic potential model for higher excited vibrational levels of the 2D stretching potential (see Fig. 1e). The main attention will be paid here to the stretching modes $(n_s, n_a)$ that form the manifold of $n = n_s + n_a$ vibrational levels. We here made the assignment $\psi_{ns,na}$ (in full agreement with a previous study[11]) assuming that the symmetry of the strict stretching wave function $\psi_{ns,na}$ is the same as in the harmonic approximation $\psi_{ns}\psi_{na}$ (see Fig. 2c–h).

When the manifold index $n$ increases, the first two levels $(n, 0)$ and $(n - 1, 1)$ becomes degenerate because of the anharmonicity[16]. Therefore, one can use the vibrational wave functions localized on the bonds on the same footing. To see this more

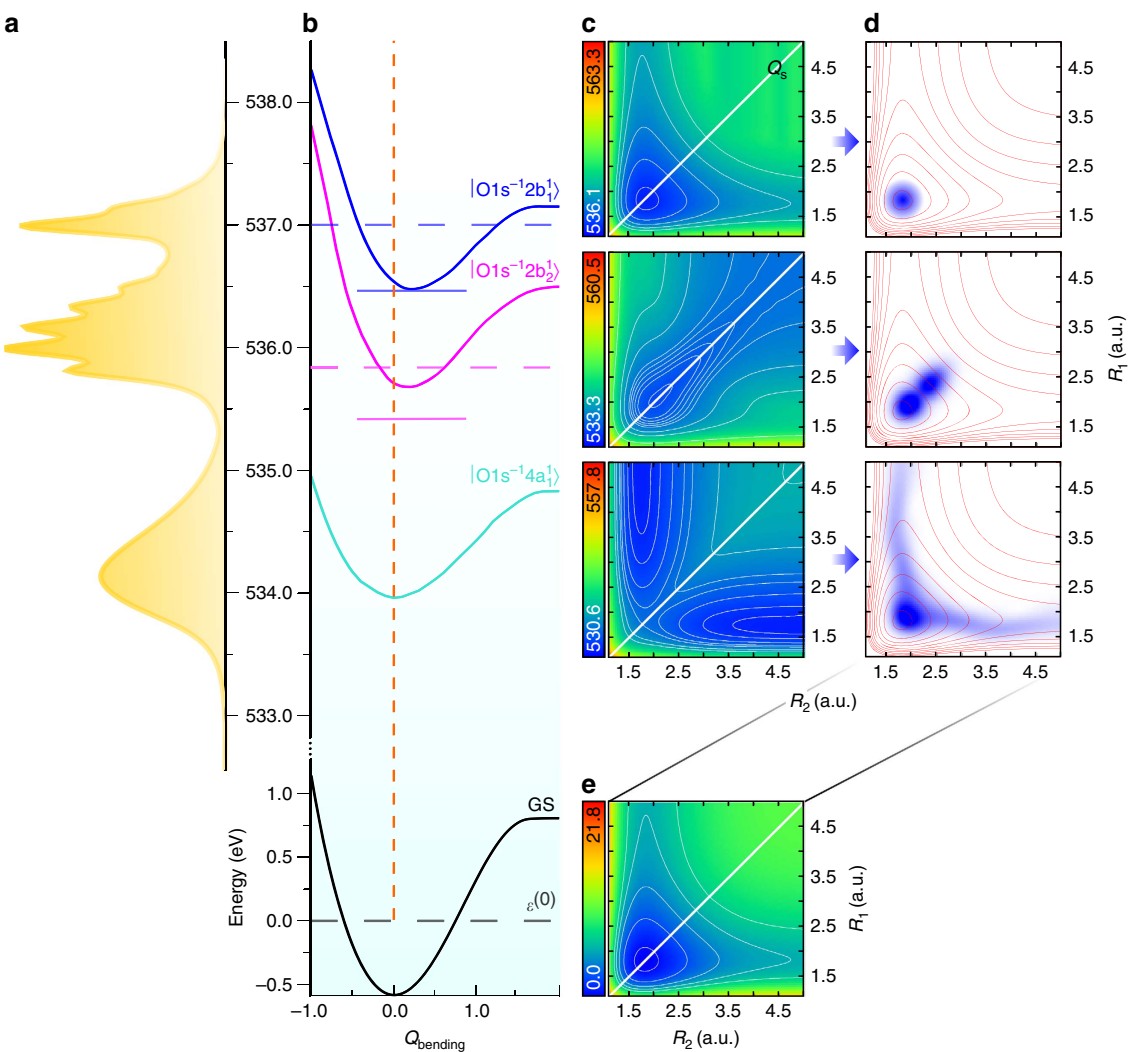

**Figure 1 | X-ray absorption spectrum and potential energy surfaces of gas-phase water. (a)** Simulated X-ray absorption spectrum for the three lowest $|O1s^{-1}4a_1^1\rangle$, $|O1s^{-1}2b_2^1\rangle$ and $|O1s^{-1}2b_1^1\rangle$ core-excited states of water. **(b)** Potential energy curves (1D) of the bending vibrational mode for the ground (GS) and core-excited states. The solid horizontal lines show the global minima of the 3D potentials, whereas the dashed horizontal lines show the position of the total zero-point energy with respect to the global minima of ground, $|O1s^{-1}2b_2^1\rangle$ and $|O1s^{-1}2b_1^1\rangle$ potential energy surfaces. The energy scale is relative to the total zero-point energy $\varepsilon^{(0)}$ (equation (1)) of ground electronic state. **(c)** Stretching potential energy surfaces (2D) as a function of bond lengths $R_1 = R_{OH_1}$ and $R_2 = R_{OH_2}$ for the core-excited states. The colour bars represent the energy range of the surfaces in eV, relative to the bottom of GS potential. $Q_s$ is the symmetric stretching coordinate. **(d)** The squared integral wave packet $|\Phi_0(0)|^2$ (see equation (2)) versus $R_1$ and $R_2$ for each of the core-excited states plotted against the contour curves of the ground-state potential for 2D stretching motion. **(e)** 2D stretching potential energy surfaces of the ground electronic state.

clearly one can construct the vibrational wave functions $\psi_1 = (\psi_{n,0} - \psi_{n-1,1})/\sqrt{2}$ and $\psi_2 = (\psi_{n,0} + \psi_{n-1,1})/\sqrt{2}$ that are exactly localized on the bond $R_1$ and $R_2$, respectively (see Fig. 2e,f). RIXS gives a unique opportunity to directly filter these localized vibrational modes by its projection onto the nuclear wave packet of the dissociative core-excited state distributed along the bonds.

**Potential energy surfaces and RIXS spectra.** Let us now demonstrate how the core excitation of nuclear motion along the reaction coordinate allows to probe the vibrational modes. This is illustrated in Fig. 1, where the PESs of the ground and core-excited states are presented together with the simulated X-ray absorption spectrum in Fig. 1a. In Fig. 1b, we notice that the bending potential is merely softened in the $|O1s^{-1}4a_1^1\rangle$

core-excited state relative to the ground state, whereas $|O1s^{-1}2b_2^1\rangle$ and $|O1s^{-1}2b_1^1\rangle$ exhibit an opening of the H-O-H angle. The $|O1s^{-1}2b_1^1\rangle$ core-excited state is of Rydberg character and has a stretching potential with a shape similar to the ground state, as seen in Fig. 1c,e. The $|O1s^{-1}4a_1^1\rangle$ PES is dissociative along the individual OH bonds, whereas in the bound $|O1s^{-1}2b_2^1\rangle$ PES there is a valley along the symmetric normal coordinate. These qualitative differences of the PESs are crucial to understand the RIXS, as it dictates/precept the wave packet propagation in the core-excited states.

Let us look on the shape of the integral wave packet of the core-excited state $|\Phi_0(0)|^2$ (2). Figure 1d shows that this wave packet is localized along the OH bonds for the dissociative $|O1s^{-1}4a_1^1\rangle$ core excitation, in full agreement with the physical picture of the dissociation along the potential valleys of this state (Fig. 1c). The picture is qualitatively different for the bound

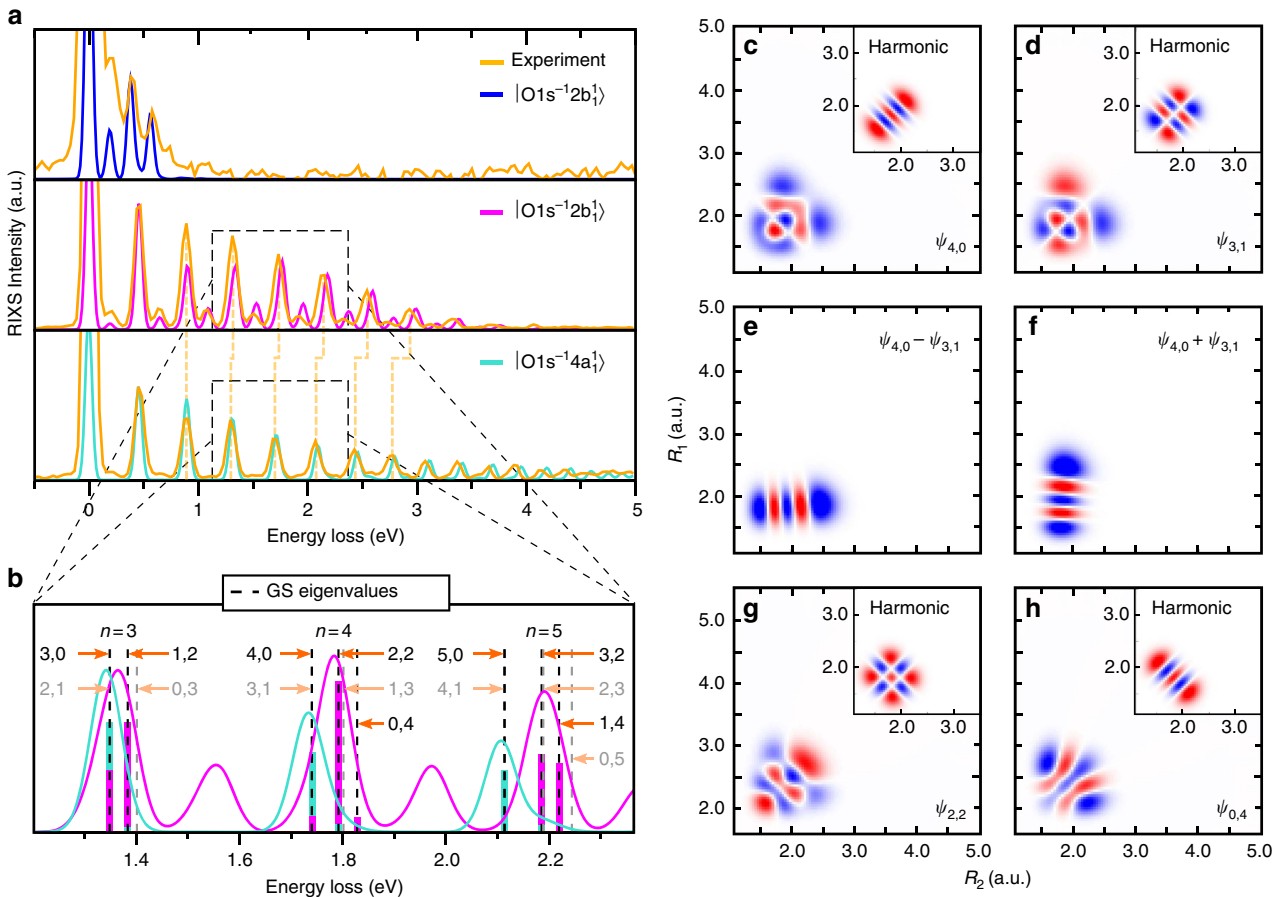

**Figure 2 | Resonant inelastic X-ray scattering spectra and vibrational wave functions.** (**a**) RIXS spectra at the $\left|O1s^{-1}2b_1^1\right\rangle$, $\left|O1s^{-1}2b_2^1\right\rangle$ and $\left|O1s^{-1}4a_1^1\right\rangle$ core-excited states obtained at detuning $\Omega = +0.20\,eV$, $-0.025\,eV$ and $+0.05\,eV$ from the top of absorption resonance[24], respectively. (**b**) Comparison between theoretical RIXS at $\left|O1s^{-1}4a_1^1\right\rangle$ and $\left|O1s^{-1}2b_2^1\right\rangle$ resonances shows the propensity rule: the final states $(n, 0)$ are suppressed at $\left|O1s^{-1}2b_2^1\right\rangle$ resonance for $n \geq 3$. The ground-state eigenvalues for $n = 3$, 4 and 5 are shown; the orange arrows point the quantum numbers $(n_s, n_a)$ that corresponds to the eigenvalues $\varepsilon_{n_s,n_a}$. (**c,d**) The degenerated vibrational wave functions $\psi_{4,0}$ and $\psi_{3,1}$ for the ground electronic state are shown. One can see that these wave functions differ qualitatively because of anharmonicity from the vibrational wave function in harmonic approximation (shown in the insets). (**e,f**) Corresponding localized vibrational states are shown. (**g,h**) The vibrational wave functions $\psi_{2,2}$ and $\psi_{0,4}$ of higher delocalized states are shown together with the corresponding wave functions in harmonic approximation.

$\left|O1s^{-1}2b_2^1\right\rangle$ core-excited state for which the potential is stretched out between the OH bonds along the symmetric coordinate $Q_s$ (Fig. 1c). This leads to $|\Phi_0(0)|^2$ being localized along $Q_s$ for the $\left|O1s^{-1}2b_2^1\right\rangle$ core excitation (Fig. 1d).

The experimental and theoretical RIXS spectra at the $\left|O1s^{-1}2b_1^1\right\rangle$, $\left|O1s^{-1}2b_2^1\right\rangle$ and $\left|O1s^{-1}4a_1^1\right\rangle$ resonances are presented in Fig. 2a. As one can notice, there are significant differences between the three spectra. First, let us analyse the RIXS via the $\left|O1s^{-1}2b_1^1\right\rangle$ resonance that displays a simpler profile of a short vibrational progression with the bending frequency. As noticed above, the 2D stretching PES of the ground and $\left|O1s^{-1}2b_1^1\right\rangle$ core-excited states (see Fig. 1c,e) are nearly parallel. As a result, one can see the quenching of the stretching vibrations in RIXS and only the bending mode is excited. More discussion of RIXS via the $\left|O1s^{-1}2b_1^1\right\rangle$ resonance can be found in Supplementary Note 4. RIXS at the $\left|O1s^{-1}4a_1^1\right\rangle$ resonance, on the contrary, does not excite the bending mode (Fig. 2a) because the bending potentials of the ground and $\left|O1s^{-1}4a_1^1\right\rangle$ core-excited states are parallel (see Fig. 1b). In this case, primarily the symmetric stretching vibrational mode is excited, as it will be discussed in more details later. As for the $\left|O1s^{-1}2b_2^1\right\rangle$ resonance, both stretching and bending motions are excited. Having only half the frequency, the bending peaks are localized between the

stretching peaks and have lower intensity (Fig. 2a). This shows clearly how different intermediate states in RIXS allow to select the vibrational excitation in the final state, thus studying a particular type of nuclear motion independently.

**Spatially selective nuclear dynamics.** Let us now focus on the stretching mode progression via the $\left|O1s^{-1}4a_1^1\right\rangle$ and $\left|O1s^{-1}2b_2^1\right\rangle$ states observed in RIXS. A closer look into the RIXS spectra of these intermediate states, shown in Fig. 2b, displays a shift between the two stretching progressions. One should notice that this shift is observed for all multiquantum $(n > 1)$ vibrational states in the full progression displayed in Supplementary Fig. 1, but it is most pronounced for $n > 3$. In order to understand this feature, we analysed the vibrational levels of the 2D ground-state potential by solving numerically the corresponding 2D eigenvalue problem (see Supplementary Note 5), and found the transition intensities from each core-excited state. The intensity of the individual vibrational resonances in RIXS is given by the squared overlap

$$\left|\left\langle \Phi_{m_c}(0)\middle|\psi_{n_s,n_a}\right\rangle\right|^2 \qquad (3)$$

between the integral wave packet in the core-excited state $\Phi_{m_c}(0)$

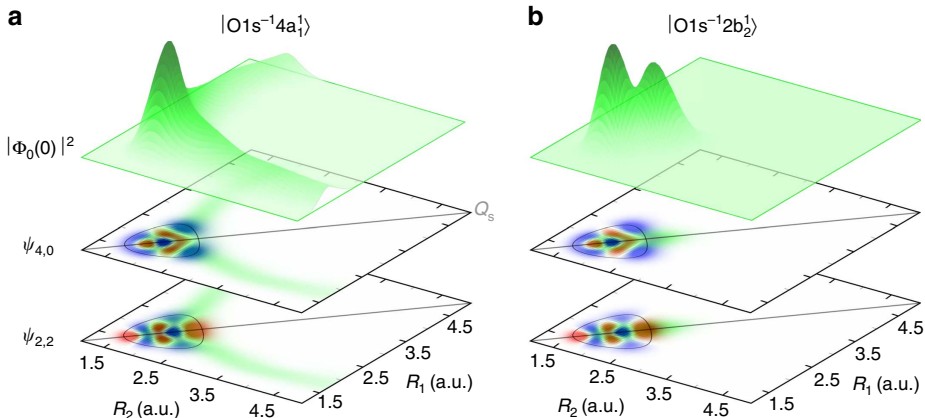

**Figure 3 | Overlap between the core-excited and ground state wave functions.** The squared integral wave packet $|\Phi_0(0)|^2$ (from Fig. 1d) versus $R_1$ and $R_2$ for the (**a**) $|O1s^{-1}4a_1^1\rangle$ and (**b**) $|O1s^{-1}2b_2^1\rangle$ core-excited states plotted against the vibrational wave functions $\psi_{4,0}$ and $\psi_{2,2}$ of higher delocalized states. Isoenergetic curves for the (4, 0) and (2, 2) vibrational states are shown with thin lines. The crossing $Q_s$ line represents the symmetric stretching coordinate.

(2) and the particular vibrational wave function $\psi_{n_s,n_a}$ of the ground electronic state. In the discussion below, we suppose that $m_c = 0$ for simplicity. As RIXS originates from the ground vibrational state $\psi_{0,0}$, only RIXS transitions to even $n_a$ antisymmetric stretching states are allowed because of the reflection symmetry of the vibrational wave functions $\psi_{n_s,n_a}(Q_s, Q_a) = (-1)^{n_a} \psi_{n_s,n_a}(Q_s, -Q_a)$. The main contribution to the line intensity (3) is defined by the maximum overlap of the core-excited wave packet distribution and lobes of the vibrational wave function $\psi_{ns,na}$. The maximum of the excited vibrational wave function is normally found near the classical turning points, where the classical speed equals zero and where the system spends most of the time[17,18]. In the 2D case studied here, the classical turning points belong to isoenergetic curves that are shown for a given vibrational level in Fig. 3.

The cross-section for RIXS to the pure symmetric stretching vibrational states $\psi_{n,0}$ is large only for the dissociative core-excited state $|O1s^{-1}4a_1^1\rangle$ where the wave packet is spread along the OH bonds. In contrast, RIXS transitions to the localized states are quenched in the case of the $|O1s^{-1}2b_2^1\rangle$ core-excited state where the wave packet is strongly confined between the bonds. From Fig. 3, it is clear that the RIXS transitions to vibrational states localized along the OH bonds ($\psi_{4,0}$ from Fig. 2c) should be strong for the $|O1s^{-1}4a_1^1\rangle$ core excitation (Fig. 3a) in contrast to the $|O1s^{-1}2b_2^1\rangle$ core-excited state (Fig. 3b) where these localized states are quenched because of negligible overlap with the wave packet $\Phi_0(0)$. However, higher vibrational states that embrace both symmetric and antisymmetric stretching excitations ($\psi_{2,2}$ from Fig. 2g) of the $n$th manifold have lobes along $Q_s$. Because of this, these states are clearly observed (Fig. 2b) in the case of the $|O1s^{-1}2b_2^1\rangle$ core excitation. This propensity rule explains the shift of the $|O1s^{-1}2b_2^1\rangle$ RIXS spectra with respect to the $|O1s^{-1}4a_1^1\rangle$ RIXS profile (Fig. 2).

## Discussion

The state-sensitive spatial localization of the integral wave packet $\Phi_0(0)$ gives us a unique tool to probe specific vibrational modes of the ground-state potential along a selected reaction pathway. Through the $|O1s^{-1}2b_1^1\rangle$ core-excited state, one can study separately the bending motion; meanwhile, the $|O1s^{-1}4a_1^1\rangle$ resonance selectively excites the symmetric stretching mode, and the core-excitation $|O1s^{-1}2b_2^1\rangle$ leads to information about the bending and a mixture of symmetric and antisymmetric stretching modes. Another possibility to study the dynamics and localization of the core-excited vibrational wave packet would be stimulated X-ray spectroscopy techniques that are under development[19,20]. However, the robust experimental realization still requires development of strong field X-ray sources in terms on stability, coherence and bandwidth.

The observed shift between the two stretching progressions has another interesting aspect: the gating effect allows to resolve fine structure within the instrumental broadening. Usually, the fine structure can be resolved only when the resolution is smaller than the spacing between resonances[21]. As one can see from Fig. 2b that each $n$th peak in the RIXS spectrum has a fine structure (see also Supplementary Fig. 1) that should be invisible because the spectral resolution (75 meV) is larger than the energy spacing between overlapping components within the $n$th manifold. In spite of this, the gating effect allows to see this fine structure via the shift of the resonant maxima. This unexpected improvement of the resolution is an important attribute of the gating effect: thanks to the propensity rules, it allows to resolve the close-lying vibrational resonances (for example, (5, 0) and (3, 2)) as they are measured separately in the two independent $|O1s^{-1}4a_1^1\rangle$ and $|O1s^{-1}2b_2^1\rangle$ RIXS spectra. Let us stress that the gating effect allows for a complete disentanglement of the vibrational modes and thus for an advanced analysis of the nuclear dynamics that cannot be achieved by simple improvement of the spectral resolution when the gating effect is absent. The gating effect is suppressed in molecular systems where the PESs of the core-excited states are similar to the ground state, that is, Rydberg series, or when they cross each other, where the vibronic coupling should be taking into account.

We have shown that different RIXS channels in $H_2O$ act as selective gates to specific vibrational modes by means of the spatially selective core-excited state dynamics. Thus, a comparison of RIXS via dissociative and bound core-excited states allows to probe vibrational modes related to different reaction pathways. This also indicates that RIXS can be used as a powerful tool to study anharmonicity of the ground-state potential by reaching highly excited vibrational levels that are not commonly accessible by conventional optical and infrared spectroscopic techniques. The spatial filtering of the final state nuclear motion can be applied for mapping of multidimensional PESs along a particular reaction coordinate. The gating effect is a general phenomenon that can be observed in many polyatomic molecules and not only in three-atomic molecules, because the generally different spatial

shapes of the multidimensional potential energy surfaces of the core-excited states result in different spatial distribution of the corresponding nuclear wave packets. Taking into account also the high element and site selectivity of RIXS, the observed phenomenon opens up potential applications for general mode localization in complex environments.

## Methods

**Experimental setup.** The experimental data were acquired using the SAXES spectrometer[22] at the RIXS end station of the ADRESS beam line[23] at the Swiss Light Source, Paul Scherrer Institut. The $H_2O(g)$ sample was prepared by evacuation and heating (60°) of a $-10$ ml $H_2O(l)$ sample reservoir. The gas was transferred towards the interaction point through previously evacuated and heated steel capillaries. The sample volume was separated from the experimental chamber by a 150 nm thick Silicon nitride membrane. A continuous sample replacement was established by constant evacuation of the $H_2O(l)$ sample reservoir and thus generating a flow of fresh sample at the interaction volume. The signal emitted from the sample volume was detected with 45° incident and emitted radiation and thus a total scattering angle of 90°. The photon energy $\omega$ used for the excitation of the sample was tuned to the resonances with the three core-excited states: $|O1s^{-1}4a_1^1\rangle$, $|O1s^{-1}2b_2^1\rangle$ and $|O1s^{-1}2b_1^1\rangle$[24]. The resonantly scattered photons were detected with a combined experimental resolution of 75 meV.

**Theoretical methods.** The PESs of the ground and core-excited states were computed with the MOLCAS 8.0 package[25] using the scalar relativistic restricted active space self-consistent field (RASSCF) method[26] followed by second-order perturbation theory (RASPT2) method[27], with the ANO-RCC[28] basis set (oxygen (14s9p4d3f2g)/[8s7p4d3f2g] and hydrogen (8s4p3d1f)/[6s4p3d1f]) in combination with a (2s2p1d) Rydberg basis in analogy with ref. 29 (see Supplementary Note 6 for more details). Ground-state normal vibrational modes were determined at the CASPT2(8, 9) level. In the RASPT2 calculations (in $C_s$ symmetry) of the potential energy surfaces, we used an active space with 10 electrons consisting of 11 orbitals in RAS2 and RAS3. Separate RASSCF calculations were performed with double or single occupation of O1s (that is frozen from the Hartree–Fock) in the RAS3 subspace to reach both ground-state and core-excited states. RASSCF state-averaging and multistate RASPT2 were performed over $|O1s^{-1}4a_1^1\rangle$ and $|O1s^{-1}2b_2^1\rangle$. All wave packet simulations were performed employing the eSPec program[30]. As $\omega$ and $\omega'$ are close to the absorption band, the self-absorption of the scattered photons was taken into account in the simulations similar to a previous study[31].

**Data availability.** The data that support the findings of this study are available from the corresponding author on reasonable request.

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

## Acknowledgements

This work was supported by the Swedish Research Council (VR), The Knut and Alice Wallenberg foundation (Grant No. KAW-2013.0020), the Carl Tryggers foundation. R.C.C. and V.V.C. acknowledge the Conselho Nacional de Desenvolvimento Científico e Tecnológico (CNPq-Brazil); F.G. and V.K. acknowledge the Russian Science Foundation (project 16-12-10109); M.D. and T.S. acknowledge the funding from the Swiss National Science Foundation within the D-A-CH programme (SNSF Research Grant 200021L 141325). S.E. and A.F. acknowledge funding from the ERC-ADG-2014 Advanced Investigator Grant no. 669531 EDAX under the Horizon 2020 EU Framework, Programme for Research and Innovation. M.O. and A.F. acknowledge partial funding by the Helmholtz Virtual Institute VI419 'Dynamic Pathways in Multidimensional Landscapes'. The calculations were performed on resources provided by the Swedish National Infrastructure for Computing (SNIC).

## Author contributions

R.C.C. performed all nuclear dynamics simulations, prepared the text of manuscript and figures; V.V.C. and F.F.G developed the software and participated in the theoretical analysis and results discussion; E.E. and M.O. did all electronic structure calculations; S.E., M.F., M.D., B.K., T.S., A.P. and A.F. suggested and planned the experiment, collected the data and carried out the data analysis; V.K., H.Å. and F.G. took main responsibility for the theoretical modelling and the writing of the paper, in which all authors contributed. All authors reviewed the manuscript.

## Additional information

**Competing financial interests:** The authors declare no competing financial interests.

**Publisher's note**: 

