## [Peer Review File · Nature Communications]

Reviewers' comments:

Reviewer #1 (Remarks to the Author):

This paper reports very high resolution RIXS spectra of water taken at 3 oxygen core excited resonances, the two valence states 4a1 and 2b2 and the Rydberg 2b1 excitations. The results highlight the power of the technique to selectively probe portions of the GS potential energy surface, and the corresponding vibrational states localization, by selecting different resonances. Beside the high quality of the experimental results, the strength of the paper is in the theoretical analysis, which is able to produce a superb simulation of the spectra, and to interpret the finest details, notably a very tiny shift, about 0.05eV judging from the figure, in the stretching vibrational progressions originating from the two valence resonances.

I think this is a high quality work, which shows the potential of the technique to explore both GS vibrational excitations and the nature of the PES in the core resonant states, also in more complex systems, and I recommend publication.

My only complaint is that (also due to the format of Nat Comm) the paper is very terse, and requires very careful reading. As an example it takes some reflection to appreciate that each "stretching" peak in fig.2 is really a manifold of sym/anti vibrational excitations. Perhaps a table or figure of the 2D computed vibrational states would help understanding. Also equilibrium values of the geometries would be worth reporting, to have a more quantitative appreciation of the change upon excitation.

- I was surprised by the O1s-2b1 spectrum on top of figure 2. Why is the exp spectrum much less resolved and noisy? And what is the reason for the irregular behavior of the vibrational intensities (0-1 is the lowest) given the simpler 1D bending)?

Reviewer #2 (Remarks to the Author):

This is a well written and timely paper that reports vibrational selectivity of spontaneous x-ray Raman signals resonant with different core transitions of water. Experiments are accompanied and supported by wavepacket simulations. The effects are attributed to the highly localized core excitations that prepare the molecule in well defined regions of configuration space

1. Details of the simulation and the 2DID model are lacking. How are the potentials and anharmonicities calculated? A supplementary information section is needed
2. Page 1 "ultra-high resolution" specify the resolution
3. There is a large body of work on stimulated x-ray Raman spectroscopy. See e.g. J Biggs, et al Ann Rev Phys Chem, 64, 101(2013) . It is different than RIXS in several respects (Time domain vs. frequency domain, stimulated vs. spontaneous, valence excitations vs. vibrational excitation) .However; the two are conceptually similar since both select excitations by the localization of the core states. Connection to that literature should be made and cited.

With these changes, I recommend that this excellent paper be accepted for publication in Nature Communications

Reviewer #3 (Remarks to the Author):

The manuscript reports on selective probing of vibrational modes (termed as selective gating by the authors) in a water molecule using soft x-ray resonant inelastic x-ray scattering.

The authors claim that their approach is quite general as it allows to break down complex multidimensional problems into elementary components.

I am not convinced that this work represents a significant advance as a universal approach to study molecular vibrations.

My reservations in recommending this paper for publication in Nature Communication are outlined below.

I would recommend to address these points and resubmit in a more specific journal.

Energy selective probing of individual vibrational states using RIXS is not a new result. It has been reported before [Hennies et al, PRL 104, 193002 (2010)] For reasons, which are unclear the authors do not mention this fact (and no relevant citations are included).

While the presented in the manuscript comprehensive interpretation of RIXS spectra of water vapor is quite interesting there is insufficient evidence that the approach can be easily applied to a wide variety of molecular systems.

The following points need to be clarified before such statements can be made:

1. In many molecular systems individual vibrational states overlap in the energy domain. In these situation selection of a particular state may require consideration of additional parameters such as X-ray polarization. Relevant limitations are not described in the manuscript.
2. The spectral resolution of RIXS of this study (75 meV) could be insufficient to selectively probe states with smaller energy separation.
3. It is unclear if the approach is suitable to all triatomic molecules or if certain limitations (such as 1 and 2) apply in particular cases.

Reply to referees reports

We are grateful to the referees for their positive consideration of our work and for encouraging us to improve our manuscript. We appreciate their constructive comments, which, we think now have been taken fully into account in the Supplementary Information and revised version of the manuscript. Please find below the detailed reply to each report. We have marked all changes made in the revised manuscript by red color.

Reply to the Reviewer #1

We are indebted to this reviewer, who acknowledges the novelty and importance of our work, by pointing out the high quality of experimental data, and recommends publication in Nature Communication.

Remark 1) My only complain is that (also due to the format of Nat Comm) the paper is very terse, and requires very careful reading. As an example it takes some reflection to appreciate that each "stretching" peak in fig.2 is really a manifold of sym/anti vibrational excitations.

In order to address this referee comment, we prepared an extended Supplementary Information which describes in more detail the methods used as well as the results. Moreover, we extended the discussion in the main text as well. Related to the referee's comment of Fig. 2, we introduced an additional Supplementary Figure 2, which clearly shows the fine structure of the vibrational resonances in RIXS.

The assignment (n_s, n_a) for each stretching peak in Figure 2b is done assuming that the symmetry of the strict stretching wave function ψ_{n_s, n_a} is similar to the wave function in the harmonic approximation $\psi_{n_s} \psi_{n_a}$. This assignment is in agreement with previous studies. We explain this point briefly in the revised text (lines 75 to 77).

Remark 2) Perhaps a table or figure of the 2D computed vibrational states would help understanding. Also equilibrium values of the geometries would be worth reporting, to have a more quantitative appreciation of the change upon excitation.

Following the suggestion of the referee we added all requested data in the corresponding section of the Supplementary Information (see Supplementary Tables 1 and 2).

Remark 3) I was surprised by the O1s-2b1 spectrum on top of Figure 2. Why is the exp spectrum much less resolved and noisy? And what is the reason for the irregular behavior of the vibrational intensities (0-1 is the lowest) given the simpler 1D bending)?

The main reason for the noisy spectrum is the rather large detuning from the narrow $2b_1$ resonance. This makes the intensity of the corresponding RIXS band smaller and hence increases noise. The main reason for the irregular behavior of the vibrational progression is the underestimation of the intensity of the zero energy loss peak (ZEL) in our theoretical simulations. As it is well established now, the intensity of this peak is affected strongly by Thomson scattering [Phys Rep 312, 87 (1999); Phys Rev Lett **106**, 153004 (2011)], which makes it stronger. The tail of the strong ZEL resonance increases the intensity of the close lying peak related to $n_b=1$, making the vibrational progression more regular, as it can be seen in the experimental profile. In our simulations we did not include the Thomson scattering as it does not affect the higher vibrational levels, we focused on in our study. Taking into account the importance of these remarks we introduced a short comment in the main

text (lines 101 and 102) and discussed this point in more detail in the Supplementary Information including corresponding references.

Reply to the Reviewer #2

We would like to equally express our gratitude to reviewer #2 who recognizes the importance of our high quality experiment and theoretical results and recommends publication in Nature Communications.

Remark 1) Details of the simulation and the 2DID model are lacking. How are the potentials and anharmonicities calculated? A supplementary information section is needed

Following the suggestion of the referee we addressed these topics in the extended Supplementary Information.

Remark 2) Page 1 “ultra-high resolution” specify the resolution

We now clarified the “ultra-high resolution” on page 1 (line 33) referring to the Methods.

Remark 3) There is a large body of work on stimulated x-ray Raman spectroscopy. See e.g. J Biggs, et al *Ann Rev Phys Chem*, 64, 101(2013) . It is different than RIXS in several respects (Time domain vs. frequency domain, stimulated vs. spontaneous, valence excitations vs. vibrational excitation). However; the two are conceptually similar since both select excitations by the localization of the core states. Connection to that literature should be made and cited.

Indeed, the stimulated RIXS process has attracted much attention thanks to the rapid development of high photon flux X-ray free-electron lasers (XFELs). Recently developed stimulated RIXS spectroscopies, addressing the dynamics of the electronic [Mukamel, et al *Ann Rev Phys Chem*, 64, 101(2013); W. Hua, et al, *Struct. Dyn.*, 3, 23601 (2016)] and nuclear [Kimberg & N. Rohringer, *Struct. Dyn.*, 3, 34101 (2016); Kimberg, et al, *Faraday Discussions*, DOI: 10.1039/C6FD00103C (2016)] wave packets, brings new possibilities for studying molecular dynamics, as compared to conventional RIXS spectroscopy. In particular, these nonlinear X-ray spectroscopic methods would allow a direct time-resolved measurements of charge transfer in molecular complexes [Schweigert & Mukamel, *Phys. Rev. A*, 76, 12504 (2007)], as well as vibrational wave packet dynamics [J. Ullrich et al, *Annu. Rev. Phys. Chem.*, 63, 635–60 (2012)]. However, the robust experimental realization of stimulated RIXS methods yet requires further development of strong field X-ray sources in terms of stability, coherence and bandwidth. When these methods are launched, they can be considered as important complementary tools for studies of the dynamics and localization of the wave packet, as discussed in our paper with the help of advanced RIXS spectroscopy.

Taking into account the importance of this remark, we include the following discussion and references in the text (lines 130 to 132):

“Another possibility to study the dynamics and localization of the core excited vibrational wave packet would be stimulated X-ray spectroscopy techniques that are under development [Mukamel, et al *Ann Rev Phys Chem*, 64, 101(2013); Kimberg & Rohringer, *Struct. Dyn.*, 3, 34101 (2016)]. However, the robust experimental realization still requires development of strong field x-ray sources in terms on stability, coherence and bandwidth.”

Reply to the Reviewer #3

We are grateful to the referee for her/his constructive comments, which we think we have fully taken into account in the revised version of the manuscript.

Remark 1) I am not convinced that this work represents a significant advance as a universal approach to study molecular vibrations.

Molecular vibrations as a means to study nuclear dynamics have drawn attention in recent years. The access to long vibrational progressions opens the possibility to probe potential energy surfaces even away from equilibrium. However, so far, the vibrational progressions of many of the larger molecules have been deemed far too complex; their multi-mode vibrations span multidimensional potential energy surfaces and disentangling these dimensions is pending. We show in this work that by using the gating effect, we can spatially select individual vibrational states in a multi-mode molecule and map, in principle, the potential energy surface along a selected reaction coordinate. To the best of our knowledge, our article presents the first experimental realization and theoretical simulation of the gating effect.

We demonstrate the spatial selection of individual vibrational states in the 3-mode showcase H_2O utilizing the state-sensitive spatial localization (in well defined directions) of the nuclear wave packet in the core-excited states. The strength of each final state vibrational resonance is defined by projection of the wave function of this vibrational level onto the core-excited nuclear wave packet. The spatial selection of the ground state vibrational levels is reached via tuning of the excitation energy to the different core-excited states with different spatial localizations of the nuclear wave packets. In particular, we have observed the discussed effect by tuning the excitation energy to the $|01s^{-1}4a_1\rangle$ and $|01s^{-1}2b_2\rangle$ core excited states with qualitatively different spatial orientations of the valleys of the 2D potentials where the corresponding wave packets are confined.

This method can in principle be adapted for other polyatomic molecules to study their ground state potentials anharmonicity and shape as well as disentangle their vibrational modes.

Remark 2) Energy selective probing of individual vibrational states using RIXS is not a new result. It has been reported before [Hennies et al, PRL 104, 193002 (2010)] For reasons, which are unclear the authors do not mention this fact (and no relevant citations are included).

We appreciate this remark because it shows concretely that our studied effect is distinct from earlier RIXS studies. While the authors of Hennies et al did probe vibrational states by selecting the excitation energy, they investigated the homonuclear diatomic molecule O_2 which only shows the $\text{O}=\text{O}$ stretch vibration, and thus cannot manifest the spatial gating effect that needs at least two vibrational modes. A single stretching vibrational progression can easily be resolved with the present day experimental setups. In our work with the more complex case of polyatomic molecules, this approach fails as the vibrational spacing is smaller than the total energy resolution. However, we still can disentangle the different vibrational modes with the help of selective gating is presented in this work. Our paper is the first experimental observation of the spatial selection of certain vibrational states in multimode molecules based on the gating effect. More precisely, the studied effect allows to select vibrational states even when the spacing between vibrational levels is smaller than the spectral resolution (see reply to remark 3b). Following the important referee remark, we included a corresponding comment in the text of the article citing the high resolution RIXS paper published by Hennies et al (lines 134 to 135).

Remark 3) While the presented in the manuscript comprehensive interpretation of RIXS spectra of water vapor is quite interesting there is insufficient evidence that the approach can be easily applied to a wide variety of molecular systems.
(The following points need to be clarified before such statements can be made:)

The state sensitive spatial orientation of the valleys of the potential energy surfaces of core-excited states is a common and general phenomenon because the shape of the potential is very sensitive to the spatial shape of the electron density. Therefore the here studied effect can be found in many polyatomic molecules and can thus be considered as a rather general phenomenon (see also reply to remark 3c). Taking into account the referee's remark we improve the correspondent discussion in the Conclusions (lines 147 to 150).

Remark 3a) In many molecular systems individual vibrational states overlap in the energy domain. In this situation selection of a particular state may require consideration of additional parameters such as X-ray polarization. Relevant limitations are not described in the manuscript.

We are grateful to the referee for this remark. The strength of a vibrational resonance in RIXS is defined by the Franck-Condon (FC) amplitude, which does not depend on the polarization \mathbf{e} in the Born-Oppenheimer approximation (BOA). The only exception is the situation when the transition dipole moment \mathbf{d} strongly depends on a molecular geometry due to the breakdown of the BOA. In this case, the factor $\mathbf{e} \cdot \mathbf{d}$ should be included in the FC amplitude. This brings the opportunity to affect the strength of vibrational transition by varying the angle between polarizations of incoming and scattered X-ray photons due to dependence of the generalized FC amplitude on the polarization. However, the dependence of \mathbf{d} on molecular geometry usually becomes significant only when the potentials of the excited electronic states cross each other or are close to each other. In the study of the H₂O molecule presented here, the BOA is valid, as the potentials of the core-excited states are nicely isolated. Due to this the polarization dependence of the vibrational profile can be ignored. Taking into account the importance of this effect we introduced a corresponding change in the main text with relevant references (lines 65 to 68).

Remark 3b) The spectral resolution of RIXS of this study (75 meV) could be insufficient to selectively probe states with smaller energy separation.

In fact this remark is related to our main result and its novelty: The quenching one of the vibrational peaks because of spatial gating effect allows to resolve vibrational peaks with the energy spacing below the spectral resolution.

To answer this question let us consider as an example the group $n=5$ (Figure 2 a,b). The selection of the vibrational states ((5,0) and (3,2) for $n=5$) is seen very clearly in the experimental data shown in Figure 2a as the shift of the maxima of the vibrational resonances in RIXS through two different core-excited states ($4a_1$ and $2b_2$). This shift is seen clearly because the (5,0) resonance is observed only in the $4a_1$ RIXS, while the (3,2) resonance is seen only in the $2b_2$ RIXS (compare panels $2b_2$ and $4a_1$ in Figure 2a). The main reason for fine resolution here is that the vibrational peaks (5,0) and (3,2) do not overlap with each other, because they are separately present in two different spectra.

The referee's comment is only true if both resonances (5,0) and (3,2) are present in the same RIXS spectrum. In this case, indeed, one cannot resolve these peaks due to the overlap of the resonances with the spacing comparable with the spectral resolution. However, since the peaks are observed in our experiment separately in two independent spectra, the observation of the spectral shift of the maxima of the peaks is not limited by the spectral resolution. The same arguments are valid for all other vibrational groups.

Taking into account the importance of this remark, we introduced a corresponding discussion in the text of the article (lines 133 to 140).

Remark 3c) It is unclear if the approach is suitable to all triatomic molecules or if certain limitations (such as 3a and 3b) apply in particular cases.

The main reason for the studied effect is the different spatial shape of the potentials of core excited states (resulting in different spatial distribution of the corresponding nuclear wave packets) which allows the spatial selection (gating) of vibrational states by tuning the excitation energy to different core-excited states.

Already from the here studied water molecule we see that all states (ground, dissociative $4a_1$ and bound $2b_2$ core-excited states) have different spatial shapes. This is a general phenomenon, in the sense that the spatial shape and orientation of multi-mode potential is very sensitive to the spatial distribution of the electron density of a given state. The discussed gating effect should be the case in many polyatomic molecules and not only in three atomic molecules. Hence, the gating in RIXS is a general principle, but the exact details are system specific because RIXS depends on the potential shapes of the specific ground and core-excited states, which forms the power of RIXS to give detailed insight into molecular electronic structure and vibrations.

The text of the article (lines 147 to 150) was improved following this referee's remark (See also reply to remarks 3a and 3b).

Reviewers' comments:

Reviewer #1 (Remarks to the Author):

The reviewer provided a confidential remarks to the editor with a recommendation to publication.

Reviewer #2 (Remarks to the Author):

The authors had responded adequately to the comments of all referees. The paper has been greatly improved. It will make a nice and novel contribution to Nature Communications. I recommend that this paper be accepted for publication

Reviewer #3 (Remarks to the Author):

The authors have addressed all comments to great detail and made a substantial effort improving the manuscript.

At this stage I can recommend the revised manuscript for publication in Nature Communications given the fact that water is a fundamentally important molecule.

I have only two minor remarks:

1. In principle, if instruments with even better spectral resolution were available the described selecting gaiting for probing vibrational modes will be simplified and possibly unnecessary in many cases.

Perhaps, further development of RIXS spectrometers should be acknowledged along with requirements on the X-ray sources.

2. Although the authors clarified many limitations of the approach it still remains unclear in which systems it would become impossible to distinguish different core-excited PES.

NCOMMS-16-14786: “Selective gating to vibrational modes through resonant X-ray scattering”

Reply to referees reports

We are grateful for all referees remarks and for their recommendation of publication. Below we address to the remarks of Reviewer #3.

Reply to the Reviewer #3

Remark 1) In principle, if instruments with even better spectral resolution were available the described selecting gating for probing vibrational modes will be simplified and possibly unnecessary in many cases. Perhaps, further development of RIXS spectrometers should be acknowledged along with requirements on the X-ray sources.

The gating effect presented in the article is related to the disentanglement of vibrational states through RIXS to different core-excited states, which is independent from the improvement of spectral resolution. In the case of $4a_1$ core-excited state, regardless of the resolution, only the vibrational states $\psi_{n,0}$ are observed in the RIXS spectrum. The benefit of higher spectral resolution would be in the RIXS at $2b_2$ resonance, which presents close resonances in each peak. With the improved resolution, this fine structure could be resolved and the precise position of these close lying resonance could be determined. In the line 145 of the article was added a comment about this.

Remark 2) Although the authors clarified many limitations of the approach it still remains unclear in which systems it would become impossible to distinguish different core-excited PES.

We can cite two situations where is impossible to distinguish different core-excited PES. The first one is when the core-excited PESs have the same shape as the ground state, which is the case of the Rydberg states. Due to the similarity of the PESs, the gating effect is absent as the stretching vibrations excitation will be quenched, which is the case of RIXS at $2b_1$ resonance. The second situation is when the core-excited PESs are close lying and the crossing between them lead to the vibronic coupling (O_2 molecule would be an example). In this case, one cannot separate the core-excited states, which also leads to the suppression of the gating effect. We add a remark about these limitations in the line 145 of the main text.